# Antibiotic use attributable to specific aetiologies of diarrhoea in children under 2 years of age in low-resource settings: a secondary analysis of the MAL-ED birth cohort

Stephanie A Brennhofer  ,[1] James A Platts-Mills,[1] Joseph A Lewnard,[2] Jie Liu,[3] Eric R Houpt,[1] Elizabeth T Rogawski McQuade  [1,4]

¹Division of Infectious Diseases and International Health, University of Virginia School of Medicine, Charlottesville, Virginia, USA
²Division of Epidemiology, University of California Berkeley, Berkeley, California, USA
³School of Public Health, Qingdao University, Qingdao, China
⁴Department of Epidemiology, Emory University, Atlanta, Georgia, USA

**Correspondence to**
Dr Elizabeth T Rogawski McQuade; erogaws@emory.edu

## ABSTRACT

**Objective** To quantify the frequency of antibiotic treatments attributable to specific enteric pathogens due to the treatment of diarrhoea among children in the first 2 years of life in low-resource settings.

**Design** Secondary analysis of a longitudinal birth cohort study, Etiology, Risk Factors, and Interactions of Enteric Infections and Malnutrition and the Consequences for Child Health and Development (MAL-ED).

**Setting** This study was conducted at eight sites in Bangladesh, Brazil, India, Nepal, Peru, Pakistan, South Africa and Tanzania.

**Participants** We analysed 9392 reported diarrhoea episodes, including 6677 with molecular diagnostic test results, as well as 31 408 non-diarrhoeal stools from 1715 children aged 0–2 years with 2 years of complete follow-up data.

**Primary and secondary outcome measures** We estimated incidence rates and the proportions of antibiotic use for diarrhoea and for all indications attributable to the top 10 aetiologies of diarrhoea. We estimated associations between specific aetiologies and antibiotic treatment, and assessed whether clinical characteristics of the diarrhoea episodes mediated these relationships.

**Results** *Shigella* and rotavirus were the leading causes of antibiotic treatment, responsible for 11.7% and 8.6% of diarrhoea treatments and 14.8 and 10.9 courses per 100 child-years, respectively. *Shigella* and rotavirus-attributable diarrhoea episodes were 46% (RR: 1.46; 95% CI: 1.33 to 1.60), and 19% (RR: 1.19; 95% CI: 1.09 to 1.31) more likely to be treated with antibiotics, respectively, compared with other aetiologies. Considering antibiotic uses for all indications, these two pathogens accounted for 5.6% of all antibiotic courses, 19.3% of all fluoroquinolone courses and 9.5% of all macrolide courses. Among indicated treatments for dysentery, *Shigella* and *Campylobacter jenjui/Campylobacter coli* were responsible for 27.5% and 8.5% of treated episodes, respectively.

**Conclusions** The evidence that *Shigella* and rotavirus were disproportionately responsible for antibiotic use due to their high burden and severity further strengthens the value of interventions targeted to these pathogens.

## Strengths and limitations of this study

► The multisite birth cohort design of this study with intensive twice-weekly home visits allowed capture of all antibiotic exposures for any indication including instances where antibiotics were obtained without prescriptions.

► The use of quantitative molecular diagnostics for a broad range of enteric pathogens allowed us to appropriately assign aetiology to diarrhoea episodes prompting antibiotic treatment.

► A limitation was that the indication for antibiotic use was not known and was therefore inferred by the overlap between treatment and diarrhoea symptoms.

Interventions against *Campylobacter* could further prevent a large burden of indicated antibiotic treatment for dysentery, which could not be averted by antibiotic stewardship interventions.

## INTRODUCTION

Diarrhoea is a major cause of antibiotic treatment among children, especially in low-income and middle-income countries (LMICs), because of both the high incidence of diarrhoea and frequency of treatment. In the multisite Etiology, Risk Factors, and Interactions of Enteric Infections and Malnutrition and the Consequences for Child Health and Development (MAL-ED) birth cohort study, the incidence of diarrhoea during the first two years of life was 273.8 episodes per 100 child years,[1] and 46% of episodes were treated with antibiotics.[2] Less than 5% of episodes were dysenteric and therefore met antibiotic treatment guidelines from the World Health Organization (WHO).[3] Nearly half of non-bloody diarrhoeal episodes were treated, representing a large burden of inappropriate

antibiotic use.[2] Similarly, in the Global Enterics Multicenter Study (GEMS), a seven-site case–control study of moderate-to-severe diarrhoea, nearly 75% of non-bloody moderate-to-severe diarrhoea episodes were treated with antibiotics among children under five.[4] Frequent antibiotic treatment of diarrhoea directly contributes to the development of antimicrobial resistance (AMR) for bacterial diarrhoeal pathogens, particularly *Shigella* and *Campylobacter*, which are on the WHO priority pathogen list for concern about AMR.[5] Treatment of diarrhoea also affects AMR more broadly through antibiotic selection pressure to bacteria carried at the time of treatment.

Because there is uncontrolled access to antibiotics in many LMICs, children often receive antibiotics without seeking care.[6] Even if a child presents to care, clinical predictors and point-of-care diagnostics to identify diarrhoea episodes that could respond to antibiotics are largely unavailable.[7] Prescribing antibiotics for diarrhoea remains the standard of care in many settings despite the recognised need for antibiotic stewardship and guidelines to reserve antibiotic treatment for dysentery.[8] Vaccines or other interventions that prevent diarrhoeal illnesses from occurring and therefore prompting treatment might provide the most effective mechanism for reducing antibiotic use.[9 10]

Influenza and pneumococcal conjugate vaccines have been found to reduce antibiotic use through the prevention of respiratory illnesses.[11] A recent randomised controlled trial demonstrated that maternal respiratory syncytial virus vaccination prevented 13% of antibiotic use in the first three months of life.[12] Additionally, rotavirus vaccination was estimated to prevent 13.6 million antibiotic-treated diarrhoea episodes annually among children under two years in LMICs.[13] Estimation of the further reductions in antibiotic use that could be achieved by vaccines against enteric pathogens such as *Shigella*, enterotoxigenic *Escherichia coli* (*E. coli*) (ETEC), *Campylobacter* and *Cryptosporidium* appropriately broadens the vaccine value proposition and could inform priority-setting for the development, evaluation and implementation of these interventions.[14]

To estimate the preventable burden of antibiotic use for diarrhoea that could be achieved by vaccines or other pathogen-specific interventions, we quantified the amount of antibiotic use that could be attributed to the treatment of specific causes of diarrhoea in the MAL-ED birth cohort study.

## METHODS
### Study design and participants
The study design for MAL-ED has been described elsewhere.[15] Briefly, this study was conducted from November 2009 to February 2014, and participants were enrolled at eight sites: Dhaka, Bangladesh; Fortaleza, Brazil; Vellore, India; Bhaktapur, Nepal; Loreto, Peru; Naushero Feroze, Pakistan; Venda, South Africa and Haydom, Tanzania. Children were followed from birth (<17 days of age)

through age 24 months. Fieldworkers conducted twice weekly home visits in which they collected information on antibiotic drug classes given to the child and diarrhoea since the last home visit. Diarrhoea was defined as three or more loose stools in a 24-hour period or visible blood in at least one stool. Diarrhoeal episodes were separated by at least two days without diarrhoea. Stool samples were collected during diarrhoea and monthly in the absence of diarrhoea. Episode severity was defined by a modified Vesikari score, previously described.[16] Dysentery was defined as reported presence of blood in at least one stool during a diarrhoeal episode. Antibiotic courses for diarrhoea were identified when antibiotic use was reported on any day during a diarrhoea episode. Distinct antibiotic courses not associated with diarrhoea were defined if separated by at least two days of no antibiotic use, as previously.[2]

### Stool testing
Pathogens were detected among all stool samples collected from children with complete follow-up. To extract total nucleic acid, the QIAamp Fast DNA Stool Mini Kit (Qiagen) was used.[17] Quantitative polymerase chain reaction (qPCR) using AgPath One Step real-time PCR kit (Thermo-Fisher) was used to detect 29 enteropathogens via the TaqMan Array Card platform.[1] A quantification cycle threshold of 35 was the analytic limit of detection. Ten enteric pathogens that were previously identified as the top causes of diarrhoea in MAL-ED[1] were included in these analyses: adenovirus 40/41, astrovirus, *Campylobacter jenjui/Campylobacter coli* (*C. jejuni/C.coli*), *Cryptosporidium*, norovirus, rotavirus, sapovirus, *Shigella*, typical enteropathogenic *E. coli* (tEPEC) and heat stable ETEC (ST-ETEC).

### Data analysis
Because multiple pathogens were frequently detected in stool during antibiotic-treated diarrhoea episodes, detection of a pathogen alone was not sufficient to assign aetiology and attribute antibiotic use. To identify the pathogens responsible for diarrhoea treated with antibiotics, we calculated pathogen-specific attributable fractions (AF) of antibiotic-treated diarrhoea using generalised linear mixed-effects models (GLMM) that associated pathogen quantity detected with presence in diarrhoeal versus non-diarrhoeal stools, as previously outlined.[1] This method leverages the quantity of pathogen detected to identify which is the most likely cause of the diarrhoea requiring treatment. The model included sex, test batch, age in quarters, pathogen quantity, pathogen quantity squared, an interaction between pathogen quantity and age, the quantity of the other nine pathogens, a random intercept for individual and a random slope for site. We calculated episode-specific pathogen AF as $AFe_i = 1\left(1/ORe_i\right)$, where $ORe$ is the pathogen-specific and quantity-specific odds ratio (OR) from the GLMM. Population-level AFs were calculated by summing the attributable fractions

per episode (AFes) across all antibiotic-treated episodes, $j$, that is, $\left(\frac{1}{j}\right) * \sum_{i=1}^{j} AFe_i$.

We calculated attributable incidence rates of antibiotic use for each pathogen per 100 child-years as the product of the AF and the total incidence of antibiotic courses for diarrhoea identified by surveillance. We also calculated the proportion of all antibiotic use that was attributable to each pathogen as the product of the AF and the proportion of all antibiotic courses that were given for diarrhoea. To quantify appropriate antibiotic use, we calculated the proportion of pathogen-attributable antibiotic use that was for dysentery. All results were stratified by age, site and antibiotic drug class.

To assess whether specific pathogens were associated with antibiotic treatment, we estimated risk ratios (RR) for the association between specific pathogens and antibiotic treatment using the pathogen-specific AFe as a continuous exposure. We used the Poisson approximation for log-binomial regression with generalised estimating equations to account for repeated episodes within each child. Estimates were scaled to represent the difference between complete attribution (AFe=1, or the maximum observed AFe for that pathogen if <1) and no attribution. Estimates were adjusted for site, age as a quadratic spline, sex and the Water, Assets, Maternal Education, Income (WAMI) index, a measure of socioeconomic status.[18]

To further assess whether diarrhoea severity mediated the associations with antibiotic treatment, we estimated the total effects of *Shigella* and rotavirus on antibiotic treatment, the pure natural direct effects (PNDE), the total natural indirect effects (TNIE) through the diarrhoea severity score and dysentery (*Shigella* only) and the proportions mediated by diarrhoea severity and dysentery using the inverse OR weighting approach to mediation analysis with weights truncated at the top 1%.[19 20] The TNIE is the magnitude of the effect of each pathogen on antibiotic use that can be explained by the association of the pathogen with diarrhoea severity, while the PNDE describes the remainder of the effect that is not mediated by severity. For the mediation analysis, aetiologies were assigned if the pathogen AFe was ≥0.5 (ie, majority attribution). For all analyses, 95% CIs were estimated by bootstrap with 1000 resamples.

### Research ethics approval statement

For the parent study, ethical approval was obtained from the Institutional Review Boards at each of the participating research sites and at the University of Virginia School of Medicine (Charlottesville, USA) (14595). For the current study, we obtained ethical approval at the University of Virginia School of Medicine (Charlottesville, USA) (22398) and Emory University (Atlanta, USA) (STUDY00003285).

### Patient and public involvement

It was not possible to involve patients or the public in the design, conduct, reporting or dissemination plans as this was a secondary data analysis of a study conducted in 2009–2014.

## RESULTS

These analyses included 1715 children with 9392 reported diarrhoeal episodes and 38 085 (n=6677 diarrhoeal, n=31 408 non-diarrhoeal) stool samples with valid qPCR results for the 10 pathogens included (table 1). Caregivers reported 15 670 antibiotic courses, among which 4335 courses were associated with treatment of diarrhoea. The overall incidence of antibiotic use due to diarrhoea was 126.38 courses per 100 child-years, and incidence was higher during the first year of life (134.11 courses per 100 child-years) than the second (118.66 courses per 100 child-years). Higher incidence in younger children reflects higher diarrhoea incidence overall, despite a lower proportion of episodes treated with antibiotics in the first year (n=2199/5015, 43.8%) compared to the second year (n=2136/4377, 48.8%). Episodes of dysentery accounted for a small proportion of diarrhoea episodes (n=461, 4.9%) and antibiotic courses for diarrhoea (n=345, 8.0%), despite the fact that 75% of dysentery episodes were treated.

*Shigella* had the highest incidence of antibiotic use of 14.77 (95% CI: 13.25 to 16.84) courses per 100 child-years, followed by rotavirus (10.90, 95% CI: 9.75 to 12.42), sapovirus (10.24, 95% CI: 8.37 to 12.55), adenovirus 40/41 (9.63, 95% CI: 8.27 to 11.69) and ST-ETEC (8.56, 95% CI: 7.04 to 10.71) (figure 1A, online supplemental table S1). *Shigella* was the leading cause of all classes of antibiotic use, except for penicillins, for which attribution was more evenly split across pathogens. Proportionally, *Shigella* and rotavirus were responsible for 11.7% (95% CI: 10.5 to 13.3) and 8.6% (95% CI: 7.7 to 9.8) of antibiotic treatments for diarrhoeal episodes, respectively (figure 2A, online supplemental table S2). These two pathogens were responsible for an even larger total proportion of fluoroquinolone (33.0%) and macrolide (28.0%) use for diarrhoea.

The amount of antibiotic use attributed to specific pathogens varied widely across sites, with more frequent pathogen-attributable use in the South Asian sites compared with African sites. *Shigella* was the leading cause of antibiotic use in India, Nepal, Peru, Pakistan and South Africa. In contrast, sapovirus was the leading cause in Brazil and Peru, adenovirus 40/41 was the leading cause in Bangladesh and ST-ETEC was the leading cause in Tanzania (online supplemental tables S3 and S4). Bangladesh was an outlier in terms of frequency; adenovirus 40/41 and *Shigella* were responsible for 50.99 (95% CI: 42.72 to 62.14) and 45.79 (95% CI: 39.70 to 54.61) courses per 100 child-years at this site alone, respectively (figure 1B; online supplemental table S5). Of note, while Pakistan had a higher incidence of antibiotic use for diarrhoea overall (373.37 per 100 child-years) than Bangladesh (213.57 per 100 child-years), many episodes in Pakistan could not be attributed to the pathogens

**Table 1** Antibiotic use, treatment of diarrhoea and stool sample collection among 1715 children enrolled in the MAL-ED cohort

| | Dhaka, Bangladesh | Fortaleza, Brazil | Vellore, India | Bhaktapur, Nepal | Loreto, Peru | Naushero Feroze, Pakistan | Venda, South Africa | Haydom, Tanzania | Overall |
|---|---|---|---|---|---|---|---|---|---|
| Children included* | 210 | 165 | 227 | 227 | 194 | 246 | 237 | 209 | 1715 |
| Total antibiotic courses | 3695 | 224 | 1740 | 1059 | 2041 | 4922 | 508 | 1481 | 15670 |
| Surveilled diarrhoeal episodes | 1520 | 168 | 960 | 1060 | 1742 | 3110 | 295 | 537 | 9392 |
| Antibiotic treatments for diarrhoea episodes (n, %)† | 897 (59.0) | 18 (10.7) | 242 (25.2) | 319 (30.1) | 688 (39.5) | 1837 (59.1) | 62 (21.0) | 272 (50.7) | 4335 (46.2) |
| Penicillin treatment (n,%)† | 133 (8.8) | 7 (4.2) | 55 (5.7) | 60 (5.7) | 150 (8.6) | 287 (9.2) | 32 (10.8) | 99 (18.4) | 823 (8.8) |
| Sulfonamide treatment (n,%)† | 2 (0.1) | 9 (5.4) | 25 (2.6) | 69 (6.5) | 195 (11.2) | 210 (6.8) | 19 (6.4) | 52 (9.7) | 581 (6.2) |
| Macrolides treatment (n,%)† | 537 (35.3) | 0 (0.0) | 11 (1.1) | 31 (2.9) | 295 (16.9) | 83 (2.7) | 2 (0.7) | 13 (2.4) | 972 (10.3) |
| Metronidazole treatment (n,%)† | 74 (4.9) | 2 (1.2) | 74 (7.7) | 161 (15.2) | 31 (1.8) | 1185 (38.1) | 6 (2.0) | 125 (23.3) | 1658 (17.7) |
| Cephalosporin treatment (n,%)† | 77 (5.1) | 1 (0.6) | 88 (9.2) | 45 (4.2) | 33 (1.9) | 575 (18.5) | 1 (0.3) | 2 (0.4) | 822 (8.8) |
| Fluoroquinolone treatment (n,%)†‡ | 252 (16.6) | 0 (0.0) | 67 (7.0) | 30 (2.8) | 72 (4.1) | 84 (2.7) | 0 (0.0) | 2 (0.4) | 507 (5.4) |
| Other antibiotic treatment (n,%)†‡ | 24 (1.6) | 0 (0.0) | 46 (4.8) | 6 (0.6) | 61 (3.5) | 792 (25.5) | 8 (2.7) | 23 (4.3) | 960 (10.2) |
| Surveilled dysentery episodes (n, %)† | 65 (4.3) | 4 (2.4) | 60 (6.2) | 48 (4.5) | 101 (5.8) | 101 (3.2) | 11 (3.7) | 71 (13.2) | 461 (4.9) |
| Antibiotic treatments for dysentery (n, %)§ | 51 (5.7) | 2 (11.1) | 27 (11.2) | 41 (12.9) | 86 (12.5) | 82 (4.5) | 4 (6.5) | 52 (19.1) | 345 (8.0) |
| Diarrhoeal stools included in the attribution analysis (n, %)† | 1379 (90.7) | 90 (53.6) | 631 (65.7) | 904 (85.3) | 1585 (91.0) | 1815 (58.4) | 115 (39.0) | 158 (29.4) | 6677 (71.1) |
| Non-diarrhoeal stools included in the attribution analysis (n, %)¶ | 3813 (84.2) | 2800 (86.4) | 4498 (88.9) | 4533 (87.8) | 3504 (81.5) | 3896 (80.0) | 4355 (80.7) | 4009 (86.1) | 31408 (84.4) |

Data are n or n (%).
Diarrhoeal and non-diarrhoeal stools included in this analysis were those that were collected and validly tested for each of the 10 pathogens.
*Children were included if they had two complete years of follow-up with qPCR data.
†N=9392.
‡N=37 216.
§N=4335.
¶Includes reported tetracyclines, other and unknown antibiotic use.
MAL-ED, Etiology, Risk Factors, and Interactions of Enteric Infections and Malnutrition and the Consequences for Child Health and Development.

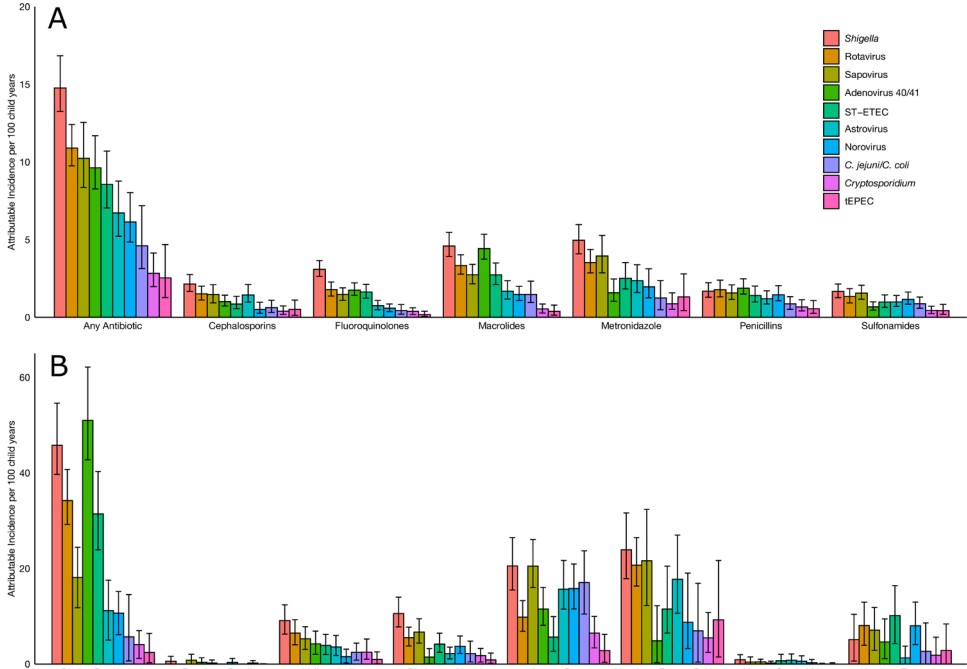

**Figure 1** Attributable incidence of pathogen-specific antibiotic courses for diarrhoea by antibiotic drug class (A) and by site (B) among 1715 children in the MAL-ED cohort. Error bars show 95% CI. *C. jejuni/C. coli*, *Campylobacter jejuni/Campylobacter coli*; MAL-ED, Etiology, Risk Factors, and Interactions of Enteric Infections and Malnutrition and the Consequences for Child Health and Development; ST-ETEC, heat-stable enterotoxigenic *Escherichia coli*; tEPEC, typical enteropathogenic *Escherichia coli*.Part of the journal style

studied. Rotavirus accounted for a lower proportion of pathogen-attributable antibiotic treatments in Brazil, Peru and South Africa compared with the other sites (online supplemental table S3).

Causes of antibiotic treatment also varied by age. In the first year of life, the pathogens responsible for the highest incidence of antibiotic treatment were rotavirus, adenovirus 40/41, sapovirus and norovirus, despite antibiotic

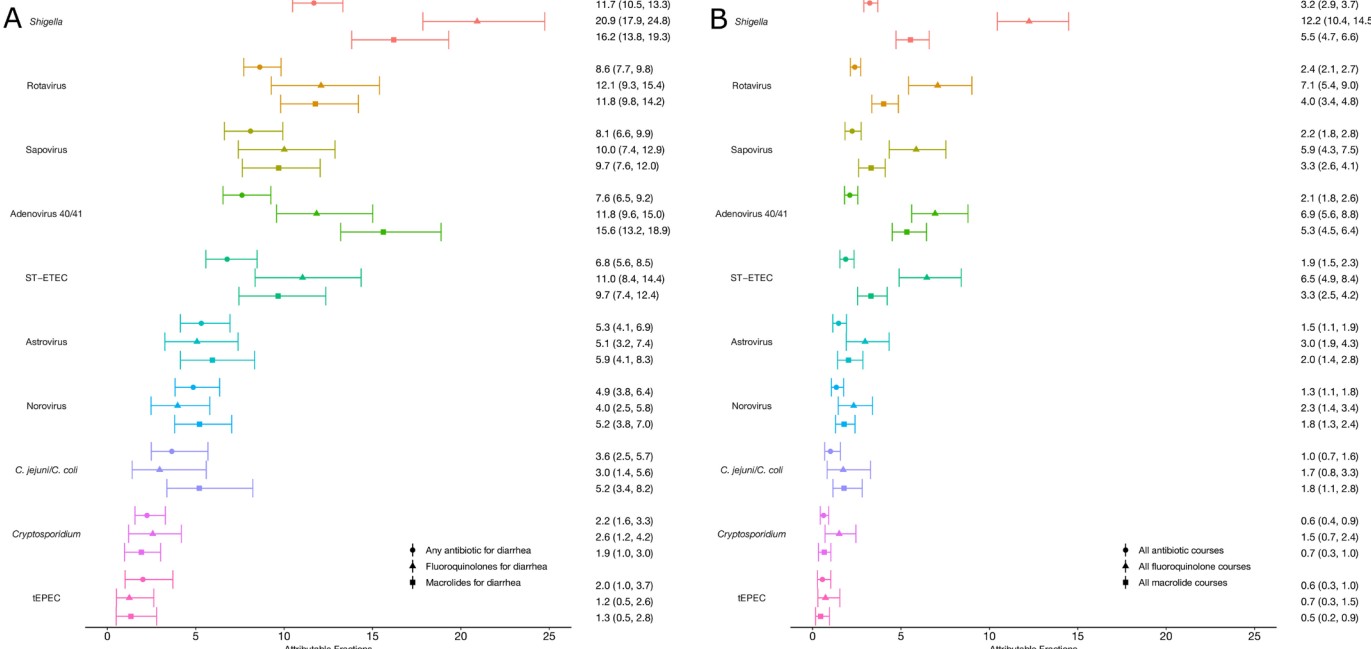

**Figure 2** Pathogen-specific attributable fractions of antibiotic courses for diarrhoea (A) and for all indications (B) by antibiotic drug class among 1715 children in the MAL-ED cohort. Error bars show 95% CI. *C. jejuni/C. coli*, *Campylobacter jejuni/Campylobacter coli*; MAL-ED, Etiology, Risk Factors and Interactions of Enteric Infections and Malnutrition and the Consequences for Child Health and Development; ST-ETEC, heat-stable enterotoxigenic *Escherichia coli*; tEPEC, typical enteropathogenic *Escherichia coli*.

use being inappropriate for the viral pathogens (online supplemental figure S1 and table S6). In the second year of life, the incidence of antibiotic use for *Shigella* was nearly twice that of any other single pathogen.

Diarrhoea was the indication for approximately one-quarter (27.7%) of antibiotic treatments overall. Therefore, specific enteric pathogens were responsible for a lower proportion of all antibiotic exposures for any indication. Overall, 3.2% and 2.4% of all antibiotic courses given were attributable to *Shigella* and rotavirus, respectively (figure 2B; online supplemental table S7). Both were responsible for a substantial proportion of treatments with specific antibiotic drug classes; 12.2% and 5.5% of fluoroquinolones and macrolides, respectively, were used for treatment of *Shigella*, and 7.1% and 4.0% of fluoroquinolones and macrolides, respectively, were used for treatment of rotavirus. All other pathogens were each responsible for approximately 2% or less of all antibiotic treatments.

Focusing on indicated antibiotic treatments, the highest proportions of antibiotic use for dysentery were attributed to *Shigella* (27.5%) and *C. jejuni/C. coli* (8.5%), respectively (online supplemental table S8). These two pathogens accounted for a larger proportion of antibiotic treated dysentery episodes compared with antibiotic treated watery diarrhoea episodes (17.2% and 5.3% more, respectively). However, less than a fifth of all antibiotic treatments attributable to *Shigella* (18.7%) and *C. jejuni/C. coli* (18.6%) were for dysentery. The AF of antibiotic treatments for dysentery compared with watery diarrhoea did not differ for the other pathogens, and less than 10% of antibiotic treatments attributed to the other pathogens were for the treatment of dysentery.

After adjustment for age, site, sex and socioeconomic status, *Shigella*-attributable diarrhoea episodes were 46% more likely to be treated with antibiotics compared with all other episodes (adjusted risk ratio (aRR): 1.46, 95% CI: 1.33 to 1.60), and rotavirus-attributable episodes were 19% more likely to be treated (1.19, 95% CI: 1.09 to 1.31) (figure 3). The associations were stronger for key drug classes; *Shigella*-attributable diarrhoea episodes were 49% more likely to be treated with fluoroquinolones or macrolides compared with other episodes (1.49, 95% CI: 1.28 to 1.73), and rotavirus-attributable episodes were 21% more likely to be treated (1.21, 95% CI: 1.04 to 1.41). The associations between *Shigella* and rotavirus and antibiotic treatment were consistent across most sites, excluding Tanzania and Nepal (online supplemental table S9). Uniquely, *Cryptosporidium* was strongly associated with antibiotic treatment in Tanzania (aRR:

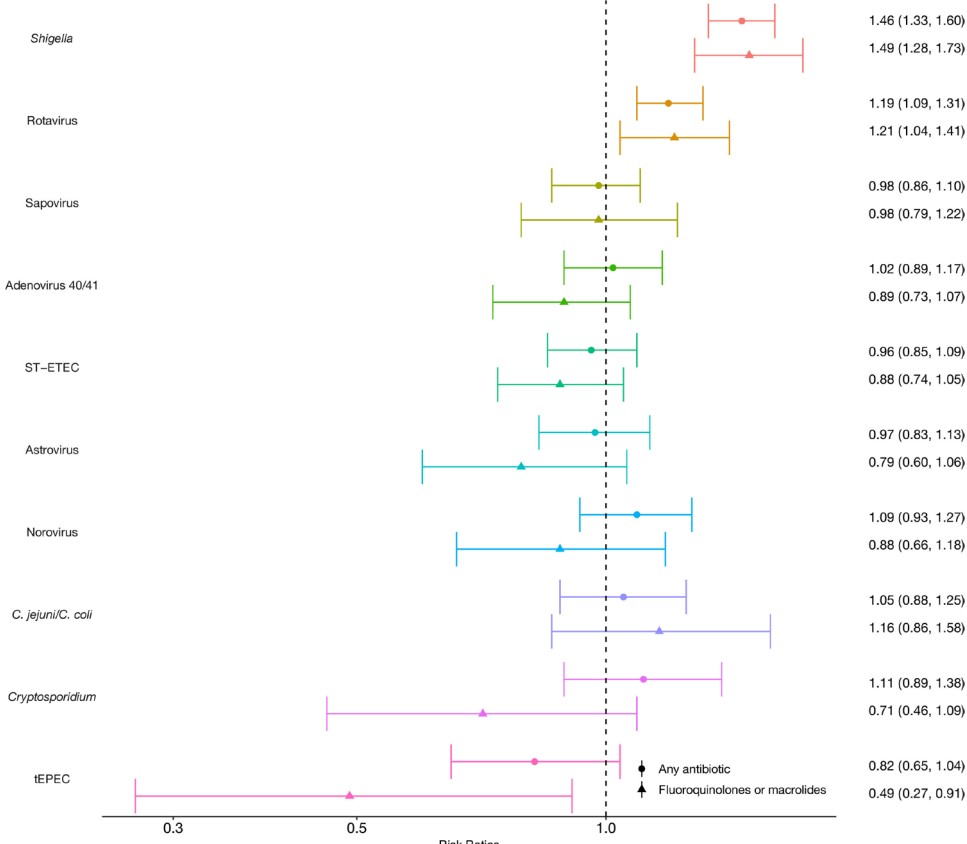

**Figure 3** Associations between specific diarrhoea aetiologies and treatment with any antibiotics and fluoroquinolones or macrolides among 1715 children in the MAL-ED cohort. Estimates are risk ratios adjusted for age, sex, socioeconomic status and site. Error bars show 95% CI. C. *jejuni/C. coli*, *Campylobacter jejuni/Campylobacter coli*; MAL-ED, Etiology, Risk Factors, and Interactions of Enteric Infections and Malnutrition and the Consequences for Child Health and Development; ST-ETEC, heat-stable enterotoxigenic *Escherichia coli*; tEPEC, typical enteropathogenic *Escherichia coli*.Part of the journal style

**Table 2** Assessment of whether diarrhoea severity and dysentery mediated the relationship between *Shigella* and rotavirus diarrhoea and antibiotic treatment among 1715 children in the MAL-ED cohort

| | *Shigella* (mediated by diarrhoea severity and dysentery) | | Rotavirus (mediated by diarrhoea severity) | |
| --- | --- | --- | --- | --- |
| | **Any antibiotic** | **Fluoroquinolones or macrolides** | **Any antibiotic** | **Fluoroquinolones or macrolides** |
| Total effect rate ratio | 1.30 (1.21, 1.40) | 1.39 (1.21, 1.58) | 1.10 (1.01, 1.19) | 1.17 (1.01, 1.33) |
| Pure natural direct effect rate ratio | 1.22 (1.12, 1.34) | 1.20 (1.02, 1.40) | 1.06 (0.97, 1.15) | 1.08 (0.89, 1.27) |
| Total natural indirect effect rate ratio | 1.07 (1.00, 1.12) | 1.16 (1.05, 1.28) | 1.04 (0.98, 1.10) | 1.08 (0.96, 1.24) |
| Proportion mediated | 0.26 (0.02, 0.50) | 0.48 (0.16, 0.90) | 0.44 (0.00, 1.00) | 0.53 (0.00, 1.00) |

Data are risk ratios (RR) with 95% CIs. The total effect rate ratio for *Shigella* and rotavirus do not equal the total effects in figure 3 as the attributable fractions per episode (AFe) were dichotomised >0.5 for the mediation models, but left continuous in figure 3.
MAL-ED, Etiology, Risk Factors, and Interactions of Enteric Infections and Malnutrition and the Consequences for Child Health and Development; .

3.18, 95% CI: 1.36 to 7.43) and India (aRR: 2.11, 95% CI: 1.18 to 3.79).

Diarrhoea severity and dysentery mediated 5% and 18% of the association between antibiotic treatment and *Shigella*, respectively (online supplemental table S10). When considered together, these two factors mediated a total 26% of the antibiotic treatment association and 48% of the fluoroquinolone and macrolide treatment association with *Shigella* (table 2). Similarly, diarrhoea severity mediated 44% of the association between rotavirus and antibiotic treatment and 53% of the association with fluoroquinolone and macrolide treatment.

## DISCUSSION

Because diarrhoea was responsible for more than a quarter of antibiotic treatments in the MAL-ED study, interventions that target specific enteric pathogens could reduce antibiotic selection pressure and make an important contribution to efforts to combat AMR. We found that *Shigella* and rotavirus were the top causes of antibiotic treatment for diarrhoea, with more than two in every 10 children on average exposed to antibiotics due to each of these pathogens in the first two years of life. Furthermore, *Shigella* was responsible for the most uses of fluoroquinolones and macrolides, which are first line therapies for *Campylobacter, Shigella* and diarrheagenic *E. coli*. While the frequency of antibiotic treatment varied by an order of magnitude across settings, *Shigella* and rotavirus were among the leading causes at all sites. Notably, rotavirus was a less frequent cause of antibiotic use in the three sites (Brazil, Peru and South Africa) that had introduced rotavirus vaccine prior to the study. Rotavirus vaccine coverage is high (>70%) and availability has expanded to all countries included in the MAL-ED study (excluding Bangladesh),[21 22] suggesting rotavirus vaccine could substantially reduce unnecessary use of antibiotics.

These results are consistent with a similar analysis of facility-ascertained moderate-to-severe diarrhoea

conducted in GEMS,[4] but have broader implications since they include antibiotic treatments for diarrhoea episodes identified in the community and therefore report much higher rates of antibiotic treated diarrhoea. In LMICs, where the majority of antibiotic use occurs outside of medically attended care, estimates of antibiotic use from healthcare settings alone are large underestimates of the total burden. This analysis also provides a broader context by considering antibiotic treatments for all indications beyond diarrhoea, which is important for LMIC settings which have high burdens of respiratory illnesses and other infections as well.

The contribution of most enteric pathogens to antibiotic use was in proportion to their contribution to diarrhoea overall. However, in addition to being the leading causes of diarrhoea in the first and second years of life, respectively, rotavirus and *Shigella* were disproportionately more responsible for antibiotic use than would have been expected based on the age-specific incidence of disease. Because point-of-care diagnostics were not available, treatment decisions were not made based on known aetiology but were rather likely due to unique features of the clinical syndromes caused by these pathogens. Indeed, we found evidence that the associations between *Shigella* and rotavirus and antibiotic treatment could be explained by the fact that these pathogens cause more severe disease. Unsurprisingly, since *Shigella* is the leading cause of dysentery for which treatment is recommended, dysentery also mediated the relationship between *Shigella* and antibiotic use. Because diarrhoea severity and dysentery only explained a portion of the relationships, there may be other subjective indicators for treatment that were insufficiently captured by the severity metrics captured.

While the contribution of individual enteric pathogens to total antibiotic use was limited (<5% for each pathogen), reductions of these magnitudes would be comparable or larger than the effect of most existing antibiotic stewardship interventions.[23] Furthermore, the attributable

proportions increased considerably for fluoroquinolones and macrolides, which are the first-line classes for diarrhoea treatment and important oral antibiotic options for a broad range of community-acquired infections. For example, *Shigella* was responsible for approximately 1 in 8 uses of fluoroquinolones and 1 in 18 uses of macrolides. *Shigella* vaccines in development[24 25] could provide an opportunity to reduce this use. Importantly, enteric viruses accounted for a quarter of all fluoroquinolone use and 16% of macrolide use. These treatment courses were not indicated and represent the burden of antibiotic overuse that could be potentially prevented by vaccines or other pathogen-specific interventions.

Interventions that reduce the incidence of bacterial diarrhoea episodes requiring antibiotics, particularly due to *Shigella* and *Campylobacter,* would also have the direct benefit of potentially preventing antibiotic-resistant disease. *Shigella* and *Campylobacter* are on the WHO priority pathogens list for research and development of new antibiotics due to increasing AMR.[26] While antibiotic resistance testing was not conducted in MAL-ED, some of the treated episodes may have been resistant to fluoroquinolones and/or macrolides, as has been reported particularly in Asia and Africa.[27–29] Specifically, a review by Gu and colleagues found that resistance to nalidixic acid and ciprofloxacin in *Shigella* spp. was 65% and 29%, respectively, in Asia and Africa in 2007–2009. Moreover, resistance rates were higher among children with diarrhoeal illnesses than adults (33.0% vs 14.3% resistance to nalidixic acid and 7.5% vs 3.6% resistance to ciprofloxacin).[27] Ghunaim *et al* found similar results regarding resistance to ciprofloxacin (fluoroquinolone) and erythromycin (macrolide) in *Campylobacter* in individuals from Asia and Africa who presented to care in Qatar. Nearly three-quarters and two-thirds of individuals from Asia and Africa, respectively, were infected with *Campylobacter* isolates resistant to ciprofloxacin, while a smaller percentage were resistant to erythromycin (7.1% in Asia vs 14.3% in Africa).[28]

Finally, because subclinical carriage of these and other bacterial enteropathogens is highly common among young children in LMICs,[30] reductions in antibiotic use overall, including treatments of viral diarrhoea, would have the important ancillary benefit of preventing antibiotic exposure to bacteria present as subclinical infections. This type of antibiotic exposure has been described as 'bystander selection', or the selective pressure for resistance on pathogens that are not the target of treatment.[31] *Shigella* and *Campylobacter* were detected in 10% and 28% of all non-diarrhoeal stools collected in MAL-ED,[30] respectively, suggesting that these pathogens were likely frequently exposed to antibiotics due to diarrhoea treatment.

Because prescriptions and/or caregiver-reported indications for treatment were unavailable, this analysis was limited by attributing antibiotic use to diarrhoea based on the temporal overlap of symptoms. Furthermore, information on specific drug given and dosing were not available, and antibiotic courses were defined based on antibiotic-free days rather than the intended duration.

The evidence that *Shigella* and rotavirus were disproportionately responsible for antibiotic use due to their high burden and severity strengthens the value proposition for rotavirus and *Shigella* vaccines[10] and other pathogen-specific interventions. These strategies could complement more generalised interventions such as educational campaigns focused on antibiotic stewardship. Prevention of diarrhoeal disease offers an important opportunity to reduce both antibiotic use and overuse.

**Contributors** SAB led data analysis, interpretation, visualisation and writing of the report. JAP-M led and conceptualised the data analysis and contributed to the interpretation, and reviewing/editing the report. JoL contributed to interpretation, and reviewing/editing the report. JiL led the development of the laboratory assays and contributed to reviewing/editing the report. ERH led funding acquisition and administration of the parent study, and contributed to reviewing/editing the report. ETRM led the conceptualisation, methodology, funding acquisition, writing of the report and contributed to data analysis, interpretation and visualisation. All authors read and approved the final manuscript. All authors accept full responsibility for the finished work and/or the conduct of the study, had access to the data, and controlled the decision to publish.

**Funding** This work was supported by Wellcome (219741/Z/19/Z to ETRM). The Etiology, Risk Factors and Interactions of Enteric Infections and Malnutrition and the Consequences for Child Health and Development Project (MAL-ED) was a collaborative project supported by the Bill & Melinda Gates Foundation (OPP1131125), the Foundation for the NIH, the National Institutes of Health, and the Fogarty International Center.

**Competing interests** None declared.

**Patient and public involvement** Patients and/or the public were not involved in the design, or conduct, or reporting or dissemination plans of this research.

**Patient consent for publication** Not required.

**Ethics approval** This study involves human participants. For the parent study, ethical approval was obtained from the Institutional Review Boards at each of the participating research sites and at the University of Virginia School of Medicine (Charlottesville, USA) (14595). For the current study, we obtained ethical approval at the University of Virginia School of Medicine (Charlottesville, USA) (22398) and Emory University (Atlanta, USA) (STUDY00003285). Participants gave informed consent to participate in the study before taking part.

**Provenance and peer review** Not commissioned; externally peer reviewed.

**Data availability statement** Data are available upon reasonable request. Deidentified participant data from the MAL-ED study is publicly available at ClinEpiDB.org after approval of a proposal by the study Pls.

**ORCID iDs**
Stephanie A Brennhofer http://orcid.org/0000-0002-9914-4471
Elizabeth T Rogawski McQuade http://orcid.org/0000-0002-4942-3747

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
