## [Reviewer comments · BMJ Open]

ARTICLE DETAILS

TITLE (PROVISIONAL)	Antibiotic use attributable to specific etiologies of diarrhea in children under two years of age in low-resource settings: a secondary analysis of the MAL-ED birth cohort
AUTHORS	Brennhofer, Stephanie; Platts-Mills, James; Lewnard, Joseph; Liu, Jie; Houpt, Eric; Rogawski McQuade, Elizabeth

VERSION 1 – REVIEW

REVIEWER	Rijkema, Sjoerd NIBSC
REVIEW RETURNED	03-Nov-2021

GENERAL COMMENTS	The authors present a thorough set of data which provides evidence that widespread, often non-targeted, antibiotic treatment of childhood diarrhoea in eight selected LMIC locations is the main cause of an increase in MDR among human pathogens in LMICs. Based on their evidence, the authors recommend that vaccination campaigns to eradicate infections of rotavirus and Shigella, being the main triggers for antibiotic treatment in children, will be an important instrument to limit the emergence of MDR strains in LMICs. And that this strategy should be pursued. This study is worthwhile of publication. SPECIFIC COMMENTS Pg 15 Ln 17: typo 'But have broader... '
--

REVIEWER	Wacharachaisurapol, Noppadol Chulalongkorn University, Pharmacology
REVIEW RETURNED	31-Jan-2022

GENERAL COMMENTS	Thank you for the opportunity to review this wonderful paper. Title: Antibiotic use attributable to specific etiologies of diarrhea in children under two years of age in low-resource settings Brennhofer et al. described antibiotic use attributable to specific etiologic causes of diarrhea in infants and young children in low-resource settings. The data were from the 7-site birth cohort with twice-weekly visits up to 2 years of follow-up. The authors also highlighted the overuse of antibiotics even though the majority of diarrhea episodes were caused by Shigella and rotavirus that are normally self-limited. Only a small number of dysentery that probably needs antibiotics treatment occurred. Even though this study was secondary data analysis, the results are essential for implementing
---

	the appropriate interventions to reduce the use and overuse of antibiotics in these particular settings. Comments Abstract  1. Would you please structure the abstract as per BMJ Open submission guidelines. 2. Page 3, lines 12-19 (Objective), the objective of the study is unclear and blended with the methodology of the study. Results  1. Were there any stool samples positive for more than one pathogen? If yes, how to select whether which one was attributable to the antibiotic use? Please give more detail (if applicable) and please add the methodology used in the part of "Methods". 2. Page 10, line 54, "Shigella and rotavirus were responsible for 11.7% (10.5-13.3) and 8.6% (7.7-9.8)...". Are the numbers in brackets 95% CI? Discussion  1. The etiologies of diarrhea were identified by PCR technic, so antibiotic susceptibility testing (AST) for bacterial etiologies could not be done. This limitation has been addressed elsewhere. However, are there any national data of the 7 study sites available especially for Shigella and Campylobacter AST on fluoroquinolones and macrolides? These data will partly help justify the use of these 2 important groups of antibiotics. 2. Page 15, lines 5-10, It is said that rotavirus was a less frequent cause of antibiotic use in the three sites that had introduced the rotavirus vaccine before this study. Are there any data on vaccine coverages on the 3 sites and also the other 4 sites? These data would highlight the benefit of the rotavirus vaccine in preventing unnecessary use of antibiotics. 3. The vaccines for preventing some high-burden pathogens such as Shigella are under development. Besides pathogen-specific interventions, in your opinion, do you think some general interventions such as education will help reduce both antibiotic use and overuse? (in the methods part, maternal education was used for adjusting the estimation of the risk ratios for the association between specific pathogens and antibiotic treatment)
--	---

VERSION 1 – AUTHOR RESPONSE

Reviewer: 1
Dr. Sjoerd Rijpkema, NIBSC

Comments to the Author:

The authors present a thorough set of data which provides evidence that widespread, often non-targeted, antibiotic treatment of childhood diarrhoea in eight selected LMIC locations is the main cause of an increase in MDR among human pathogens in LMICs. Based on their evidence, the authors recommend that vaccination campaigns to eradicate infections of rotavirus and Shigella, being the main triggers for antibiotic treatment in children, will be an important instrument to limit the emergence of MDR strains in LMICs. And that this strategy should be pursued. This study is worthwhile of publication.

SPECIFIC COMMENTS

- Pg 15 Ln 17: typo 'But have broader...'

Thank you for catching this typo. We have replaced the period after "GEMS" with a comma.

****This sentence now reads: "These results are consistent with a similar analysis of facility-ascertained moderate-to-severe diarrhea conducted in GEMS,⁴ but have broader implications since they include antibiotic treatments for diarrhea episodes identified in the community and therefore report much higher rates of antibiotic treated diarrhea."**

Reviewer: 2

Dr. Noppadol Wacharachaisurapol, Chulalongkorn University

Comments to the Author:

Thank you for the opportunity to review this wonderful paper.

Title: Antibiotic use attributable to specific etiologies of diarrhea in children under two years of age in low-resource settings

Brennhofner et al. described antibiotic use attributable to specific etiologic causes of diarrhea in infants and young children in low-resource settings. The data were from the 7-site birth cohort with twice-weekly visits up to 2 years of follow-up. The authors also highlighted the overuse of antibiotics even though the majority of diarrhea episodes were caused by Shigella and rotavirus that are normally self-limited. Only a small number of dysentery that probably needs antibiotics treatment occurred. Even though this study was secondary data analysis, the results are essential for implementing the appropriate interventions to reduce the use and overuse of antibiotics in these particular settings.

Comments

Abstract

1. Would you please structure the abstract as per BMJ Open submission guidelines.

We have revised our abstract to have the following headings: objective, design, setting, primary and secondary outcome measures, results, and conclusions. We did not include the "intervention" section as there was no intervention in our study.

2. Page 3, lines 12-19 (Objective), the objective of the study is unclear and blended with the methodology of the study.

**** The objective section has been revised for clarity and now reads, "To quantify the frequency of antibiotic treatments attributable to specific enteric pathogens due to the treatment of diarrhea among children in the first two years of life in low-resource settings."**

Results

1. Were there any stool samples positive for more than one pathogen? If yes, how to select whether which one was attributable to the antibiotic use? Please give more detail (if applicable) and please add the methodology used in the part of "Methods".

Yes, many stools were positive for more than one pathogen. We used the attributable fraction methodology previously used in MAL-ED and other studies to attribute etiology. This method is described in the methods section and involves associating pathogen quantity with diarrhea to identify episodes in which the pathogen was detected at a high enough quantity to be associated with diarrhea (and is therefore deemed the etiologic pathogen). We have added several more clarifying and contextual sentences to the methods section to highlight where these methods are discussed.

****We have added the following to the data analysis section: “Because multiple pathogens were frequently detected in stool during antibiotic-treated diarrhea episodes, detection of a pathogen alone was not sufficient to assign etiology and attribute antibiotic use.” ... “This method leverages the quantity of pathogen detected to identify which is the most likely cause of the diarrhea requiring treatment.”**

2. Page 10, line 54, “Shigella and rotavirus were responsible for 11.7% (10.5-13.3) and 8.6% (7.7-9.8)...”. Are the numbers in brackets 95% CI?

****We have clarified the numbers within parentheses to indicate they are 95% CI, which now reads, “Proportionally, *Shigella* and rotavirus were responsible for 11.7% (95% CI: 10.5-13.3) and 8.6% (95% CI: 7.7-9.8) of antibiotic treatments for diarrheal episodes, respectively (Figure 2A, Table S2).”**

Discussion

1. The etiologies of diarrhea were identified by PCR technic, so antibiotic susceptibility testing (AST) for bacterial etiologies could not be done. This limitation has been addressed elsewhere. However, are there any national data of the 7 study sites available especially for *Shigella* and *Campylobacter* AST on fluoroquinolones and macrolides? These data will partly help justify the use of these 2 important groups of antibiotics.

Thank you for this comment. We were not aware of any national data on AST for fluorquinolones and macrolides for *Shigella* and *Campylobacter* specifically. We have added more details on the available data on resistance prevalence for these pathogens in the discussion section.

****We have added the following to the discussion section, “Specifically, a review by Gu and colleagues found that resistance to nalidixic acid and ciprofloxacin in *Shigella* spp. was 65% and 29%, respectively, in Asia and Africa in 2007-2009. Moreover, resistance rates were higher amongst children with diarrheal illnesses than adults (33.0% vs. 14.3% resistance to nalidixic acid and 7.5% vs. 3.6% resistance to ciprofloxacin).²⁷ Ghunaim et al. found similar results regarding resistance to ciprofloxacin (fluoroquinolone) and erythromycin (macrolide) in *Campylobacter* in individuals from Asia and Africa who presented to care in Qatar. Nearly three-quarters and two-thirds of individuals from Asia and Africa, respectively, were infected with *Campylobacter* isolates resistant to ciprofloxacin, while a smaller percentage were resistant to erythromycin (7.1% in Asia vs. 14.3% in Africa).²⁸”**

2. Page 15, lines 5-10, It is said that rotavirus was a less frequent cause of antibiotic use in the three sites that had introduced the rotavirus vaccine before this study. Are there any data on vaccine coverages on the 3 sites and also the other 4 sites? These data would highlight the benefit of the rotavirus vaccine in preventing unnecessary use of antibiotics.

We agree that this would be helpful data. Country-level data is available publicly, though we do not have vaccine coverage data at the specific study sites. We have added these details to the discussion.

****We have indicated the three countries which had rotavirus vaccine prior to the start of the study. “Notably, rotavirus was a less frequent cause of antibiotic use in the three sites (Brazil, Peru, and South Africa) that had introduced rotavirus vaccine prior to the study.”**

****We have added the following to the discussion section as an update as to where the other 4 countries are at in terms of rotavirus vaccination status, “Rotavirus vaccine coverage is high (>70%) and availability has expanded to all countries included in the MAL-ED study (excluding Bangladesh),^{21,22} suggesting rotavirus vaccine could substantially reduce unnecessary use of antibiotics.”**

3. The vaccines for preventing some high-burden pathogens such as *Shigella* are under development. Besides pathogen-specific interventions, in your opinion, do you think some general interventions such as education will help reduce both antibiotic use and overuse? (in the methods part, maternal

education was used for adjusting the estimation of the risk ratios for the association between specific pathogens and antibiotic treatment)

Thank you for this comment. Other interventions focused on antibiotic stewardship such as education campaigns could potentially reduce antibiotic overuse and we have added this comment to the discussion section. We previously noted in the discussion that existing antibiotic stewardship interventions have had limited effectiveness, however. Maternal education was included as an adjustment covariate for the risk ratios between pathogens and antibiotic treatment as a broad marker of educational achievement and socioeconomic status; it was not specific to education related to antibiotics and therefore is not a good marker of whether educational interventions could reduce antibiotic use.

****We have added the following sentence to the final paragraph of the discussion, “These strategies could complement more generalized interventions such as educational campaigns focused on antibiotic stewardship.”**

VERSION 2 – REVIEW

REVIEWER	Wacharachaisurapol, Noppadol Chulalongkorn University, Pharmacology
REVIEW RETURNED	09-Mar-2022
GENERAL COMMENTS	I would like to thank the authors for their clarification. I have no other comments on this manuscript.